# Metabolite profiling and bioactivity of *Cicerbita alpina* (L.) Wallr. (Asteraceae, Cichorieae)

**DOI:** 10.3390/plants12051009

**Published:** 2023-02-23

**Authors:** Dimitrina Zheleva-Dimitrova, Alexandra Petrova, Gokhan Zengin, Kouadio Ibrahime Sinan, Vessela Balabanova, Olivier Joubert, Christian Zidorn, Yulian Voynikov, Rumyana Simeonova, Reneta Gevrenova

**Affiliations:** 1Department of Pharmacognosy, Faculty of Pharmacy, Medical University-Sofia, 1000 Sofia, Bulgaria; 2Department of Pharmacology, Pharmacotherapy and Toxicology, Faculty of Pharmacy, Medical University-Sofia, 1000 Sofia, Bulgaria; 3Department of Biology, Faculty of Science, Selcuk University, Campus, 42250 Konya, Turkey; 4Institut Jean Lamour, UMR CNRS 7198, Université de Lorraine, CNRS, IJL, F-54000 Nancy, France; 5Pharmazeutisches Institut, Abteilung Pharmazeutische Biologie, Christian-Albrechts-Universität zu Kiel, 24118 Kiel, Germany; 6Department of Chemistry, Faculty of Pharmacy, Medical University-Sofia, 1000 Sofia, Bulgaria

**Keywords:** *Cicerbita alpina*, secondary metabolites, antioxidant properties, enzyme inhibitory activity, cytotoxic activity, UHPLC-HRMS

## Abstract

*Cicerbita alpina* (L.) Wallr. is a perennial herbaceous plant in the tribe Cichorieae (Lactuceae), Asteraceae family, distributed in the mountainous regions in Europe. In this study, we focused on the metabolite profiling and the bioactivity of *C. alpina* leaves and flowering heads methanol-aqueous extracts. The antioxidant activity of extracts, as well as inhibitory potential towards selected enzymes, involving in several human diseases, including metabolic syndrome (α-glucosidase, α-amylase, and lipase), Alzheimer’s disease, (cholinesterases: AChE, BchE), hyperpigmentation (tyrosinase), and cytotoxicity were assessed. The workflow comprised ultra-high-performance liquid chromatography—high-resolution mass spectrometry (UHPLC-HRMS). UHPLC-HRMS analysis revealed more than 100 secondary metabolites, including acylquinic, acyltartaric acids, flavonoids, bitter sesquiterpene lactones (STLs), such as lactucin, dihydrolactucin, their derivatives, and coumarins. Leaves showed a stronger antioxidant activity compared to flowering heads, as well as lipase (4.75 ± 0.21 mg OE/g), AchE (1.98 ± 0.02 mg GALAE/g), BchE (0.74 ± 0.06 mg GALAE/g), and tyrosinase (49.87 ± 3.19 mg KAE/g) inhibitory potential. Flowering heads showed the highest activity against α-glucosidase (1.05 ± 0.17 mmol ACAE/g) and α-amylase (0.47 ± 0.03). The obtained results highlighted *C. alpina* as a rich source of acylquinic, acyltartaric acids, flavonoids, and STLs with significant bioactivity, and therefore the taxon could be considered as a potential candidate for the development of health-promoting applications.

## 1. Introduction

*Cicerbita alpina* (L.) Wallr. (*Lactuca alpina* (L.) A.Gray) (alpine chicory, blue sow thistle) is a perennial herbaceous plant in the tribe Cichorieae (Lactuceae), Asteraceae family, and is distributed in the mountainous regions in Europe [1]. Commonly, its edible shoots are used as a vegetable or for salads [2]. As a vegetable, the species has a commercial value, and recently, some trials for cultivation have been performed [3,4]. 

Based on the literature survey of the secondary metabolites, the species contains sesquiterpene lactones and sesquiterpene lactone glucosides, which are also used as chemosystematic/chemophenetic markers for the tribe [5]. Roots/subaerial parts contain the guaianolides 11*β*,13-dihydrolactucin, 8-acetyl-lactucin, 8-acetyl-11*β*,13-dihydrolactucin, and lactucin, 8-acetyl-15*β*-D-glucopyranosyllactucin, and the non-guaiane type sesquiterpene sonchuside A [6,7,8]. Additionally, furanocoumarins (imperatorin, isoimperatorin, oxypeucedanin, and ostruthol) were detected in the roots [9]. The shoots of alpine chicory are characterized by phenolic acids such as chlorogenic acid, 3,5-dicaffeoylquinic acid, caftaric acid, and mostly cichoric acid [2]. 

The usage of traditionally consumed wild edible plants of the genus *Lactuca*, which expands dietary variety, is declining, and their potential health-promoting effect is undervalued [10]. In comparison to lettuce (*Lactuca sativa* L.) and its cultivars, wild lettuces are still poorly studied chemically. As the wild relatives constitute a gene pool for the popular vegetable, their chemical composition could be of importance for breeders. One of the most characteristic features of the *Lactuca* species examined so far is their ability to synthesize bitter sesquiterpene lactones of different structural types, including guaianolides, germacranolides, and eudesmanolides [5,11]. The presence of sesquiterpene lactones, e.g., the guaianolide lactucin and its derivatives, are responsible for the anti-inflammatory, analgesic, and sedative activities [12] as well as for the bitter taste and the antifeedant activity of the species [13,14]. 

The presence of phenolics in plant species, e.g., caffeic acid derivatives, reveals the radical scavenging potential and UV radiation protection [15,16]. Moreover, the occurrence of cichoric acid is related to a variety of potential health benefits such as antioxidant, anti-inflammatory, obesity prevention, and neuroprotection effects [17]. Previous investigation revealed a large amount of antioxidant caffeic acid derivatives in the edible shoots of *C. alpina*. Assessed in the DPPH assay, the dicaffeoyl derivatives cichoric acid and cynarin (1,5-dicaffeoylquinic acid) had the higher free radical scavenging potential of related monocaffeoyl derivatives, caffeoyltartaric acid (caftaric acid) and chlorogenic acid, respectively [18].

Despite the studies on the chemical composition of *C. alpina* so far, there is no in-depth investigation of both secondary metabolites and the biological potential of the species. Detailed information on the metabolite profile in leaves and flowering heads would provide valuable information on the biofunctional potential of these compounds. Herein, the majority of annotated acylquinic acids, acyltartaric acids, and flavonoids were reported for the first time in *C. alpina*. Moreover, this is the initial report on the chemical composition of alpine chicory flowering heads. Further investigation aims at the cultivation of *C. alpina* as a potential candidate for attenuating metabolic-related disorders with nutraceutical and pharmaceutical applications. 

## 2. Results and Discussion 

### 2.1. UHPLC-HRMS Profiling of Specialized Metabolites in C. alpina Extracts

Based on the retention times, MS and MS/MS accurate masses, fragmentation patterns in MS/MS spectra, relative ion abundance, and comparison with reference standards and literature data, 110 specialized natural products were identified or tentatively annotated in *C. alpina* extracts (Table 1). The total ion chromatograms (TIC) of the studied extracts in negative ion mode are depicted in Appendix A. 

#### 2.1.1. Carboxylic, Hydroxybenzoic and Hydroxycinnamic Acids

Hydroxybenzoic acids (**1**, **5**, **6**, **9**, **11**, **13**), their glycosides (**2**–**4**, **7**–**8**), quinic (**10**), shikimic acid (**12**), hydroxycinnamic acids (**16**, **18**, **19**, **20**), and their glycosides (**14**–**15**, **17**) were identified based on comparison of retention times, exact masses and fragment spectra with reference standards and literature data (Table 1) [19]. Compounds **21** [M−H]^−^ at *m*/*z* 295.046 and **22** [M−H]^−^ at *m*/*z* 309.060 gave base peaks at *m*/*z* 133.012 [C_4_H_5_O_5_]^−^ and 147.028 [C_5_H_7_O_5_]^−^, respectively, corresponding to the loss of a caffeoyl residue (−162.033 Da). The subsequent loss of H_2_O (−18 Da) and CO_2_ (−44 Da) in the fragmentation pathways of the aforementioned compounds indicated the presence of hydroxyl and carboxyl groups. MS/MS spectrum of **22** was similar to that of **21**, except for the appearance of one methylene group. Thus, **21** and **22** were identified as caffeoyl-malic acid and caffeoyl-methylmalic acid, respectively (Table 1).

#### 2.1.2. Acylquinic Acids

A variety of acylquinic acids (AQA), including six mono-, eleven di-, and three triacylquinic acids, were identified/annotated in *C. alpina* extracts (Table 1, Appendix A). Fragmentation patterns and diagnostic ions in the MS/MS spectra of AQA were reported elsewhere [19,20,21]. 

Three isobars shared the same deprotonated molecule [M−H]^−^ at *m*/*z* 353.086; the main compound **25** was assigned to chlorogenic acid (5-caffeylquinic acid) by the base peak at *m*/*z* 191.055 [quinic acid-H]^−^. The positional isomer neochlorogenic acid (3-caffeylquinic acid) (**23**) was discernable by the higher relative abundances of the fragment ions at *m*/*z* 179.033 (58.6%) and 135.043 (51.0%) than those of **25**. Compounds **23** and **25** were also unambiguously identified by comparison with reference standards. In the MS/MS spectrum of **26**, a base peak at *m*/*z* 173.044 [quinic acid-H-H_2_O]^−.^was observed, indicating a caffeoyl residue at position 4 of the quinic acid. Thus, **26** was annotated as 4-caffeylquinic acid [20].

The compound 3-*p*-coumaroylquinic acid (3-*p*-CoQA) (**24**) was deduced from the base peak at *m*/*z* 163.039 [*p*-CoA-H]^−^ (Table 1). Compounds **27** and **29** showed a base peak at *m*/*z* 191.055 and fragment ions at *m*/*z* 163.038 [*p*-CoA-H]- and 193.050 [FA-H]-, respectively, and were assigned to 5-*p*-coumaroylquinic and 5-feruloylquinic (FQA) acids [19,20,22,23] (Table 1). 

Four compounds (**34**–**37**) shared the same deprotonated molecule at *m*/*z* 515.119 together with the distinctive fragment ions for dicaffeylquinic acids (diCQA) at *m*/*z* 353.087 [M−H-caffeoyl]^−^, 335.076 [M−H-caffeoyl-H_2_O]^−^, 191.055 [QA-H]^−^, 179.0339 [CA-H]^−^, and 173.044 [QA-H-H_2_O]^−^. Compounds were identified based on the different intensities of above mention ions and comparison with Clifford’s hierarchical key [19,20,22,23] (Table 1). The presence of an abundant ion at *m*/*z* 173.045 in the MS/MS spectra indicated vicinal diCQA (**34** and **37**). Compound **34** gave diagnostic fragment ion at *m*/*z* 335.077 [CQA-H-H_2_O]^−^ along with *m*/*z* 135.044 as observed for 3,4-diCQA [19]. On the other hand, the low abundant “dehydrated” ion at *m*/*z* 335 indicated 4,5-diCQA (**37**). Regarding compound **35**, the abundant ion at *m*/*z* 191.055 (99.3%) and the absence of the peak at *m*/*z* 335.077 indicated 3,5-diCQA, while **36** was ascribed as 1,5-diCQA. Compounds **38**, **41**, and **42** shared the same deprotonated molecules at *m*/*z* 499.125 (C_25_H_23_O_11_), together with fragment ions characteristic for *p*-coumaric acid at *m*/*z* 337.093 [*p*-CoQA-H]^−^, *m*/*z* 163.039 [*p*-CoA-H], and *m*/*z* 119.049 [*p*-CoA-H-CO_2_]^−^. The compounds were annotated as *p*-coumaroyl-caffeoylquinic acids (Table 1). Similarly to the MS/MS fragmentation patterns of diCQA and F-CQA, compounds **28**, **30**, and **31** were assigned to hydroxydihydrocaffeoyl-caffeoylquinic acids (HC-CQA) [19,24].

Compounds **32** and **33** yielded a precursor ion at *m*/*z* 677.173 (consistent with C_31_H_33_O_17_) accompanied by the relevant fragment ions at *m*/*z* 515.121 [M−H-Hex]^−^, 353.088 [M−H-Hex-caffeoyl]^−^ and 191.055 [M−H-Hex-2caffeoyl]^−^ indicating subsequent losses of a hexose unit and two caffeoyl residues. The 3,4-diCQA core (**32**) was deduced from the ions at *m*/*z* 173.044 (100%) and 179.034 (74.7%), and 135.044 (61.8%). Accordingly, **32** was ascribed as 3,4-dicaffeoylquinic acid-hexoside. In the same manner, 1,3-dicaffeoylquinic acid-hexoside (**33**) was discernable by the ions at *m*/*z* 135.044 (100%), 179.034 (97.2%), and 341.088 (53.7%).

Compound **43** showed a deprotonated molecule at *m*/*z* 677.152, together with relevant fragment ions at *m*/*z* 515.120 [M−H-caffeoyl]^−^, 353.088 [M−H-2caffeoyl]^−^, and 191.055 [M−H-3caffeoyl]^−^, indicative for triCQA. The 3,4,5-tricaffeoylquinic acid is evidenced by the abundant fragment ions at *m*/*z* 173.044 (94%), 179.034 (72.9%), and 135.044 (77.0%) [21]. 

#### 2.1.3. Acyltartaric Acids

Similarly to acylquinic acids, a variety of acyltartaric acids (ATA) was annotated, including two mono-ATA, seven di-ATA, and three triacyltartaric acids (Table 1). MonoATA were deduced from the prominent ions at *m*/*z* 149.008 [TA-H]^−^ (tartaric acid, TA) supported by *m*/*z* 112.984 [TA-H-2H_2_O]^−^ and 103.002 [TA-H-H_2_O-CO]^−^. Within this group, caffeoyltartaric (**44**) and *p*-coumaroyltartaric acid (**49**) were found. Compounds **47** and **48** were consistent with dicaffeoyltartaric acids affording prominent fragment ions at *m*/*z* 311.041 (83.6%) and 149.008 (100%) (Figure 1). 

The assignment of two isobars of feruloyl-caffeoyltartaric acids **52** and **55** (at *m*/*z* 487.088) was confirmed by the fragments at *m*/*z* 325.057 [M−H-caffeoyl]^−^ and 293.031 [M−H-ferulic acid]^−^. Ferulic acid was also deduced from the fragment ions at *m*/*z* 193.050 [ferulic acid-H]^−^ and 134.036 [ferulic acid-H-CO_2_-CH_3_]^−^, while caffeic acid was evidenced by *m*/*z* 179.038 [caffeic acid-H]^−^, 161.023 [caffeic acid-H-H_2_O]^−^, and 135.044 [caffeic acid-H-CO_2_]^−^. 

In the same way, three *p*-coumaroyl-caffeoyltartaric acid isomers **50**, **53,** and **54** were annotated at *m*/*z* 457.077 [M−H]^−^ supported by the prominent ions at *m*/*z* 295.046 [M−H-caffeoyl]^−^, 163.039 [*p*-coumaric acid-H]^−^, and 119.049 [*p*-coumaric acid-H-CO_2_]^−^. MS/MS spectra of three tricaffeoyltartaric acids (**45**, **46,** and **51**) were acquired. They afforded a precursor ion at *m*/*z* 635.106 together with the transitions at *m*/*z* 635.106 → 473.073 → 293.031 → 149.009 resulting from the losses of three caffeoyl residues. This class of secondary metabolites shows a significant degree of stereoisomerism [25].

##### Flavones, Flavonols, and Flavanones

A key step in the dereplication/annotation of flavonoid glycosides was the neutral loss of 162.05, 146.05, 132.04, 176.03, and 204.06 Da, corresponding to hexose, deoxyhexose, pentose, hexuronic acid, and acetylhexose [26]. A series of ions in (−) ESI/MS/MS from neutral losses of CH_2_O (−30 Da), C_2_H_2_O (−42 Da), CO (−28 Da), CO_2_ (−44 Da), H_2_O (−18 Da), (CH_2_O + CO) (−58 Da), (H_2_O + CO) (−46 Da), (2CO) (−56 Da), and (CO_2_ + CO) (−72 Da) were used for the dereplication of the flavonoid aglycones [27]. 

Fragment ions resulting from the retro-Diels–Alder (RDA) reaction of the flavonoid skeleton are informative. Ions formed after C-ring cleavage of the aglycon are presented as ^i,j^A–/^i,j^B– (in negative mode) [27].

##### Flavones

In the MS/MS spectra of compounds **73**, **77**, **79**, **81**, **87**, and **88**, the aglycone was identified as the flavone apigenin (**93**) based on fragment ions at *m*/*z* 239.036 [Api-H-CH_2_O]^−^, 211.040 [Api-H-CH_2_O-CO]^−^, together with RDA ions ^1,3^B^−^ at *m*/*z* 117.033, ^1,3^A^−^ at *m*/*z* 151.003, and ^0,4^A^−^ at *m*/*z* 107.012. Luteolin (**91**) and ten of its glycosides (**63**–**65**, **68**, **71**–**72**, **80**, **82**–**83**, and **89**) were also identified. The identification of the aglycone luteolin was determined based on a series of fragment ions at *m*/*z* 285.041 [Lu-H]^−^, 255.030 [Lu-H-CH_2_O]^−^, 257.042 [Lu-H-CO]^−^, 241.051 [Lu-H-CO_2_]^−^, 227.034 [Lu-H-CH_2_O-CO]^−^, 211.039 [Lu-H-H_2_O-2CO]^−^, together with RDA ions ^1,3^B^−^ at *m*/*z* 133.029, ^1,3^A^−^ at *m*/*z* 151.003, and ^0,4^A^−^ at *m*/*z* 107.012 (Table 1). 

In the MS/MS spectra of **83** and **88**, a consecutive loss of dihydroxybutyryl (C_4_H_6_O_3_) (−102.031 Da), acetyl (C_2_H_2_O) (−42.011 Da), and hexosyl (C_6_H_10_O_5_) (−162.053) radicals were observed. Thus, the compounds were annotated as 7-hydroxybutyryl-*O*-acetylhexosides of luteolin (**83**) and apigenin (**88**), respectively (Table 1, Appendix A).

**Table 1 plants-12-01009-t001:** Specialized metabolites in *Cicerbita alpina* extracts.

№	Identified/Tentatively Annotated Compound	Molecular Formula	Exact Mass [M − H]^−^	Fragmentation Pattern in (−) ESI-MS/MS	t_R_ (min)	Δ ppm	Distribution	Level of Identification [28]
**Carboxylic, hydroxybenzoic acids, and their glycosides**								
**1**	gallic acid ^a^	C_7_H_6_O_5_	169.0131	169.0131 (35.1), 125.0228 (100), 97.0278 (4.2), 79.0173 (1.0)	1.15	6.843	1,2	1
**2**	gallic acid-*O*-hexoside	C_13_H_16_O_10_	331.0671	331.0671 (100), 169.0128 (2.9), 168.0052 (34.6), 149.9945 (15.0), 125.0229 (29.5),	1.25	0.212	1,2	2
**3**	syringic acid-*O*-hexoside	C_15_H_20_O_10_	359.0985	359.0985 (44.7), 197.0446 (100), 179.0340 (23.4), 135.0437 (32.9), 123.0436 (21.5), 107.0484 (0.9)	1.52	0.473	1,2	2
**4**	protocatechuic acid-*O*-hexoside	C_13_H_16_O_9_	315.0722	315.0721 (100), 153.0180 (0.3), 152.0101 (58.4), 109.0286 (9.0), 108.0200 (85.0), 81.0329 (0.8)	1.75	0.048	1,2	2
**5**	vanillic acid ^a^	C_8_H_8_O_4_	167.0350	167.0338 (25.5), 152.0103 (7.4), 149.0230 (37.9), 123.0071 (100), 108.0201 (31.4)	2.01	−1.232	1,2	1
**6**	protocatechuic acid ^a^	C_7_H_6_O_4_	153.0182	153.0180 (16.9), 109.0279 (100), 91.0172 (0.5), 81.0328 (1.4)	2.06	8.966	1,2	1
**7**	protocatechuic acid-*O*-hexoside isomer	C_13_H_16_O_9_	315.0722	315.0730 (35.6), 153.0180 (100), 135.0438 (0.4),123.0435 (56.4), 109.0278 (27.4)	2.17	2.681	1,2	2
**8**	4-hydroxybenzoic acid-*O*-hexoside	C_13_H_16_O_8_	299.0772	299.0772 (14.5), 137.0229 (100), 109.0281 (0.6)	2.49	0.136	1,2	2
**9**	4-hydroxybenzoic acid ^a^	C_7_H_6_O_3_	137.0244	137.0228 (100), 108.0200 (7.0), 93.0328 (2.9)	2.86	11.804	1,2	1
**10**	quinic acid	C_7_H_12_O_6_	191.0561	191.0550 (100), 173.0444 (1.8), 155.0340 (0.4), 127.0386 (3.3), 111.0435 (1.6), 93.0329 (6.0), 85.0278 (19.7)	3.21	5.712	1,2	2
**11**	gentisic acid ^a^	C_7_H_6_O_4_	153.0182	153.0179 (76.4), 135.0073 (29.2), 109.0279 (100), 91.0172 (3.7), 81.0328 (1.1)	3.86	9.293	1,2	1
**12**	shikimic acid	C_7_H_10_O_5_	173.0455	173.0451 (100), 144.0443 (16.6), 93.0329 (68.6)	6.24	5.991	1,2	2
**13**	salicylic acid ^a^	C_7_H_6_O_3_	137.0244	137.0230 (34.6), 109.0278 (3.2), 108.0200 (3.5), 93.0330 (100)	6.29	10.417	1,2	1
**Hydroxycinnamic acids and derivatives**								
**14**	caffeic acid-*O*-hexoside	C_15_H_18_O_9_	341.0867	341.0879 (4.8), 179.0338 (100), 135.0437 (58.7), 123.0436 (0.5), 107.0488 (0.7)	2.44	0.160	1,2	2
**15**	caffeoylgluconic acid	C_15_H_18_O_10_	357.0827	357.0826 (24.7), 339.0728 (11.3), 195.0501 (100), 179.0339 (13.6), 177.0392 (18.8), 161.0233 (4.5), 135.0437 (22.6), 129.0178 (11.5), 87.0070 (10.8)	2.77	0.392	1,2	2
**16**	rosmarinic acid ^a^	C_18_H_16_O_8_	359.0767	359.0769 (80.7), 315.0886 (9.8), 161.0595 (100), (65.1), 135.0286 (2.6)	3.15	0.865	1	1
**17**	caffeic acid-*O*-hexoside isomer	C_15_H_18_O_9_	341.0867	341.0878 (10.9), 179.0338 (100), 135.0436 (71.3), 107.0486 (1.0)	3.29	0.104	1,2	2
**18**	*p*-coumaric acid ^a^	C_9_H_8_O_3_	163.0389	163.0388 (8.0), 135.0437 (0.7), 119.0485 (100)	3.36	7.590	1,2	1
**19**	caffeic acid ^a^	C_9_H_8_O_4_	179.0338	179.0338 (20.5), 135.0436 (100), 117.0333 (0.6), 107.0486 (1.3)	3.55	6.602	1,2	1
**20**	ferulic aci ^a^	C_10_H_10_O_4_	193.0493	193.0493 (1.4), 178.0259 (25.5), 149.0598 (2.0), 134.0358 (100)	3.56	1.332	1,2	1
**21**	caffeoylmalic acid	C_13_H_12_O_8_	295.0464	295.0464 (0.3), 179.0339 (13.0), 161.0229 (0.3), 135.0437 (9.0), 133.0124 (100), 115.0021 (27.2), 89.0227 (1.7), 72.9914 (2.6), 71.0122 (10.3)	4.15	1.422	1,2	2
**22**	caffeoylcitramalic acid	C_14_H_14_O_8_	309.0607	309.0607 (0.5), 179.0304 (4.7), 161.0231 (6.4), 147.0284 (100), 135.0437 (3.7), 129.0178 (39.8), 101.0228 (7.2), 87.0071 (1.0), 85.0279 (9.2)	4.77	2.979	1,2	2
**Acylquinic acids**								
**23**	neochlorogenic (3-caffeoylquinic) acid ^a^	C_16_H_18_O_9_	353.0867	353.0878 (40.7), 191.0551 (100), 179.0338 (58.6), 173.0442 (3.9), 161.0229 (1.8), 135.0436 (51.0), 127.0389 (1.5), 111.0432 (1.7), 93.0327 (4.1), 85.0277(8.1)	2.35	0.015	1,2	1
**24**	3-*p*-coumaroylquinic acid	C_16_H_18_O_8_	337.0929	337.0912 (8.8), 173.0447 (1.7), 163.0388 (100), 135.0436 (0.8), 119.0487 (29.8), 93.0329 (2.8), 85.0277 (0.6)	3.04	5.042	1	2
**25**	chlorogenic (5-caffeoylquinic) acid ^a^	C_16_H_18_O_9_	353.0867	353.0878 (4.8), 191.0550 (100), 179.0340 (1.0), 173.0445 (0.7), 161.0233 (1.5), 135.0435 (0.9), 127.0386 (1.5), 111.0434 (1.0), 93.0330 (2.8), 85.0279 (7.9)	3.21	0.070	1,2	1
**26**	4-caffeoylquinic acid	C_16_H_18_O_9_	353.0867	353.0880 (28.1), 191.0551 (54.0), 179.0339 (65.7), 173.0443 (100), 161.0233 (3.2), 135.0437 (53.9), 127.0385 (2.2), 111.0436 (3.8), 93.0329 (23.0), 85.0278 (10.2)	3.38	0.495	1,2	2
**27**	5-*p*-coumaroylquinic acid	C_16_H_18_O_8_	337.0929	337.0931 (10.5), 191.0550 (100), 173.0444 (7.2), 163.0388 (5.3), 135.0433 (0.2), 127.0386 (1.0), 119.0486 (5.0), 111.0436 (2.4), 93.0329 (16.8), 85.0279 (4.8)	3.95	0.651	1,2	2
**28**	3-caffeoyl-5-hydroxydihydrocaffeoylquinic acid	C_25_H_26_O_13_	533.1303	533.1303 (81.2), 371.0981 (10.4), 353.877 (19.4), 335.0773 (4.9), 191.0551 (100), 179.0339 (49.5), 173.0443 (15.7), 161.0231 (7.0), 135.0437 (60.5), 111.0436 (2.2), 93.0330 (9.0), 85.0279 (7.2)	4.02	0.462	1,2	2
**29**	5-feruloylquinic acid	C_17_H_20_O_9_	367.1035	367.1035 (16.7), 191.0550 (100), 173.0444 (14.1), 135.0391 (0.2), 134.0358 (11.3), 127.0384 (1.2), 111.0436 (4.0), 93.0329 (26.2), 85.0279 (5.5)	4.41	0.015	1,2	2
**30**	1/3/5-caffeoyl-4-hydroxydihydrocaffeoylquinic acid	C_25_H_26_O_13_	533.1306	533.1306 (100), 371.1010 (9.3), 353.0878 (5.9), 335.0779 (10.3), 191.0550 (22.5), 179.0339 (22.4), 173.0444 (75.7), 161.0230 (15.9), 135.0436 (71.7), 111.0433 (4.4), 93.0330 (20.8), 85.0276 (1.8)	4.45	0.912	1,2	2
**31**	1-caffeoyl-3-hydroxydihydrocaffeoylquinic acid	C_25_H_26_O_13_	533.1305	533.1305 (32.1), 371.0985 (54.9), 353.0892 (4.0), 335.0781 (3.2), 197.0449 (4.3), 191.0552 (15.5), 179.0340 (11.6), 173.0444 (19.1), 161.0235 (4.2), 135.0437 (100), 123.0437 (2.4), 93.0330 (5.4), 85.0278 (1.5)	4.55	0.912	1	2
**32**	1,3,4-tricaffeoylquinic acid	C_31_H_34_O_17_	677.1533	677.1533 (79.4), 515.1213 (38.0), 353.0876 (22.3), 335.0804 (0.5), 191.0549 (11.8), 179.0338 (74.7), 173.0444 (100), 135.0437 (61.8), 93.0329 (30.0), 85.0276 (1.9)	5.11	3.170	1	2
**33**	1,3,5- tricaffeoylquinic acid	C_31_H_34_O_17_	677.1528	677.1528 (93.1), 515.1365 (41.5), 353.0878 (11.8), 341.0877 (53.7), 335.0748 (1.0), 191.0550 (15.5), 179.0339 (97.2), 173.0444 (55.1), 161.0233 (6.4), 135.0437 (100), 93.0329 (25.3), 85.0275 (3.7)	5.68	2.358	1,2	1
**34**	3,4-dicaffeoylquinic acid ^a^	C_25_H_24_O_12_	515.1195	515.1200 (100), 353.0878 (15.4), 335.0768 (7.2), 191.0551 (32.2), 179.0339 (55.8), 173.0444 (63.7), 161.0230 (14.7), 135.0437 (45.3), 111.0436 (4.1), 93.0329 (16.1), 85.0278 (2.0)	5.69	0.972	1,2	1
**35**	3,5-dicaffeoylquinic acid ^a^	C_25_H_24_O_12_	515.1195	515.1193 (17.6), 353.0879 (100), 191.0551 (99.3), 179.0338 (47.6), 173.0445(4.4), 161.0232 (4.5), 135.0437 (63.3), 111.0434 (2.9), 93.0329 (4.1), 85.0279 (11.1)	5.89	0.465	1,2	1
**36**	1,5-dicaffeoylquinic acid ^a^	C_25_H_24_O_12_	515.1195	515.1200 (36.9), 353.0879 (80.6), 335.0774 (1.5), 191.0551 (100), 179.0339 (50.5), 173.0448 (9.5), 135.0438 (51.0), 127.0388 (0.9), 111.0435 (3.2), 93.0328 (5.1), 85.0279 (7.9)	6.03	−0.465	1,2	1
**37**	4,5-dicaffeoylquinic acid ^a^	C_25_H_24_O_12_	515.1195	515.1201 (80.6), 353.0880 (52.4), 335.0794 (0.6), 191.0551 (32.7), 179.0339 (65.4), 173.0443 (100), 135.0436 (59.7), 127.0383 (1.4), 111.0435 (3.9), 93.0329 (24.5), 85.0277 (4.5)	6.24	0.465	1,2	1
**38**	3-*p*-coumaroyl-5-caffeoylquinic acid	C_25_H_24_O_11_	499.1246	499.1256 (17.7), 353.0882 (5.4), 337.0932 (65.5), 191.0552 (8.2), 173.0443 (7.7), 163.0388 (100.0), 161.0231 (2.5), 135.0436 (2.8), 119.0487 (36.8), 111.0436 (1.5), 93.0331 (2.8), 85.0277 (0.7)	6.52	1.994	1,2	2
**39**	3-feruloyl-5-caffeoylquinic acid	C_26_H_26_O_12_	529.1351	529.1362 (12.5), 367.1035 (84.8), 335.0779 (0.7), 193.0496 (100), 191.0552 (3.7), 179.0337 (0.5), 173.0443 (8.0), 161.0232 (7.4), 135.0386 (3.2), 134.0358 (61.1), 111.0437 (1.5), 93.0331 (2.5), 85.0278 (0.5)	6.81	1.929	1,2	2
**40**	1,3-dicaffeoyl-5-hydroxydihydrocaffeoylquinic acid	C_34_H_32_O_16_	695.1616	695.1616 (73.0), 533.1310 (7.4), 515.1198 (7.1), 353.0775 (9.6), 191.0551 (100), 179.0339 (36.7), 173.0443 (15.8), 135.0437 (58.3), 111.0432 (1.3), 93.0327 (7.5), 85.0278 (6.6)	6.84	0.227	1	2
**41**	4-*p*-coumaroyl-5-caffeoylquinic acid	C_25_H_24_O_11_	499.1246	499.1251 (28.9), 353.0870 (0.5), 337.0931 (62.3), 191.0552 (3.2), 173.0443 (100), 163.0388 (15.6), 135.0436 (1.2), 119.0486 (0.4), 111.0437 (3.7), 93.0329 (22.5), 85.0277 (0.5)	6.93	0.952	1,2	2
**42**	4-caffeoyl-5-*p*-coumaroylquinic acid	C_25_H_24_O_11_	499.1246	499.1249 (100), 353.0875 (70.4), 337.0931 (8.9), 191.0552 (69.6), 179.0327 (70.6), 173.0444 (91.5), 161.0237 (73.8), 135.0437 (73.8), 119.4812 (2.4), 93.0331 (23.3), 85.0279 (4.8)	7.63	0.952	1,2	2
**43**	3,4,5-tricaffeoylquinic acid	C_34_H_30_O_15_	677.1512	677.1521 (3.40). 515.1195 (31.2), 353.0875 (45.6), 335.0771 (17.8), 191.0549 (43.9), 179.0339 (72.9), 173.0443 (94.0), 161.0230 (28.1), 135.0437 (77.0), 111.0438 (5.1), 93.0329 (21.2)	7.78	1.339	1,2	2
**Acyltartaric acids**								
**44**	caftaric acid	C_13_H_12_O_9_	311.0411	311.0411 (0.4), 179.0338 (71.1), 149.0077 (100), 135.0436 (49.3), 112.9864 (1.3), 103.0020 (1.6), 87.0071 (13.0)	2.39	0.819	1,2	2
**45**	tricaffeoyltartatric acid	C_31_H_24_O_15_	635.1064	635.1064 (19.6), 473.0940 (41.5), 455.0835 (7.9), 341.0881 (11.4), 293.0304 (11.8), 219.0292 (7.0), 179.0338 (100), 161.0228 (4.0), 149.0076 (0.8), 135.0436 (75.4), 112.9865 (8.2), 103.0006 (0.4), 87.0073 (3.2)	3.87	3.396	1,2	2
**46**	tricaffeoyltartatric acid	C_31_H_24_O_15_	635.1065	635.1065 (23.0), 473.0940 (65.6), 455.0833 (10.2), 341.0877 (51.3), 293.0304 (36.1), 219.0293 (16.7), 179.0338 (98.8), 161.0231 (12.8), 149.0078 (14.0), 135.0436 (100), 112.9865 (23.9), 87.0071 (8.1)	4.33	3.491	1,2	2
**47**	dicaffeoyltartaric acid (cichoric acid)	C_22_H_18_O_12_	473.07	473.0726 (6.6), 311.0410 (83.6), 293.0304 (20.7), 179.0338 (80.3), 149.0077 (100), 135.0436 (71.6), 161.0230 (3.8), 112.9865 (8.2), 103.0021 (2.6), 87.0071 (14.8)	4.95	0.171	1,2	2
**48**	dicaffeoyltartaric acid (cichoric acid)	C_22_H_18_O_12_	473.07	473.0740 (6.2), 311.0410 (92.9), 293.0305 (18.1), 179.0338 (53.1), 161.0233 (4.4), 149.0078 (100), 135.0437 (44.8), 112.9865 (11.5), 103.0021 (2.4), 87.0072 (16.7)	5.25	0.210	1,2	2
**49**	*p*-coumaryltartaric acid	C_13_H_12_O_8_	295.0447	295.0447 (1.8), 163.0387 (100), 133.0491 (5.3), 119.0486 (29.4), 149.0076 (0.8), 112.9864 (7.7), 103.0020 (2.4), 87.0070 (6.5)	5.71	4.272	1,2	2
**50**	caffeoyl-p-coumaroyltartaric caid	C_22_H_18_O_11_	457.0777	457.0777 (3.1), 295.0459 (86.4), 277.0352 (37.9), 231.0298 (3.6), 219.0294 (24.9), 179.0339 (47.6), 203.0341 (20.1), 163.0388 (100), 149.0078 (10.2), 135.0437 (55.0), 119.0486 (52.5), 112.9864 (38.0), 103.0022 (2.8), 87.0071 (14.7)	5.73	0.143	1,2	2
**51**	tricaffeoyltartatric acid	C_31_H_24_O_15_	635.1056	635.1056 (29.6), 473.0729 (93.8), 341.0668 (77.5), 323.0569 (3.0), 293.0305 (37.9), 219.0293 (13.5), 179.0339 (17.9), 161.0232 (14.6), 149.0083 (2.5), 145.0282 (20.9), 135.0437 (28.4), 112.9865 (100), 87.0072 (8.2)	6.01	2.587	1	2
**52**	caffeoyl-feruloyltartaric acid	C_23_H_20_O_12_	487.0881	487.0881 (5.1), 325.0566 (100), 307.0463 (35.2), 293.0305 (65.1), 233.0450 (17.3), 219.0293 (32.3), 193.0497 (87.2), 179.0338 (72.7), 161.0230 (20.1), 135.0437 (76.7), 134.0358 (60.8), 112.9864 (52.5), 103.0020 (4.3), 87.0071 (16.5)	6.09	0.142	1,2	2
**53**	caffeoyl-p-coumaroyltartaric acid isomer	C_22_H_18_O_11_	457.0778	295.0459 (100), 277.0354 (17.2), 219.0292 (17.2), 203.0345 (7.3), 179.0338 (1.9), 163.0388 (85.0), 149.0076 (35.3), 135.0436 (39.0), 119.0488 (39.8), 112.9864 (31.7), 103.0022 (2.9), 87.0071 (14.2)	6.11	0.275	1,2	2
**54**	caffeoyl-p-coumaroyltartaric acid isomer	C_22_H_18_O_11_	457.0771	457.0771 (5.8), 295.0456 (97.0), 277.0349 (13.2), 219.0290 (15.6), 203.0348 (10.3), 179.0338 (35.1), 163.0388 (95.9), 161.0235 (9.8), 149.0076 (100), 135.0436 (48.9), 119.0486 (51.2), 112.9864 (40.3), 103.0021 (6.3), 87.0070 (25.7)	6.67	1.060	1	2
**55**	caffeoyl-feruloyltartaric acid isomer	C_23_H_20_O_12_	487.0885	487.0885 (47.4), 325.0566 (77.9), 293.0303 (2.7), 179.0339 (9.3), 163.0235 (100), 161.0231 (52.3), 135.0437 (23.0), 134.0362 (1.1), 112.9865 (4.2), 103.0020 (19.0), 87.0070 (3.5)	7.87	0.556	1,2	2
**Flavones, flavonols, and flavanones**								
**56**	quercetin 3-*O*-hexosyl-*O*-hexuronide	C_27_H_28_O_18_	639.1212	639.1212 (100), 463.0887 (79.8), 343.0458 (1.0), 301.0349 (23.2), 300.0276 (44.3), 271.0247 (46.5), 178.9987 (3.5), 151.0025 (5.6), 107.0125 (0.8)	3.26	1.460	2	2
**57**	quercetin 3,7-*O*-dihexoside	C_27_H_30_O_17_	625.1418	625.1418 (100), 463.0876 (21.7), 462.0809 (35.1), 301.0353 (36.0), 300.0256 (7.8), 271.0247 (45.3), 151.0023 (5.7), 121.0282 (0.6), 107.0122 (1.7)	3.40	1.212	2	2
**58**	isoetin *O*-hexosyl-*O*-hexoside	C_27_H_30_O_17_	625.1414	625.1414 (100), 463.0892 (0.7), 301.0353 (66.7), 300.0274 (17.9), 271.0243 (0.5), 243.0294 (0.8), 151.0024 (3.4), 149.0231 (3.9), 107.0122 (2.2)	4.01	1.708	2	3
**59**	apigenin 7-*O*-hexosyl-*O*-hexuronide	C_27_H_28_O_16_	607.1315	607.1315 (100), 445.0759 (1.8), 431.0985 (21.1), 345.0608 (0.4), 311.0579 (2.8), 269.0454 (64.5), 268.0377 (41.4), 151.0027 (2.0), 117.0177 (1.9), 107.0124 (0.9)	4.32	1.766	2	2
**60**	isoetin-7-*O*-hexoside	C_21_H_20_O_12_	463.0887	463.0887 (100), 301.0351 (36.2), 300.0276 (34.7), 271.0242 (0.7), 227.0348 (0.7), 243.0291 (6.6), 151.0022 (7.6), 149.0442 (6.3), 149.0236 (0.9), 107.0120 (3.5)	4.55	0.455	2	3
**61**	isorhamnetin 3,7-*O*-dihexoside	C_28_H_32_O_17_	639.1577	639.1577 (100), 578.2167 (1.2), 519.1140 (0.7), 477.1035 (6.3), 476.0967 (15.3), 315.0513 (9.8), 314.0408 (9.4), 313.0356 (56.9), 300.256 (0.7), 285.0405 (9.0), 151.0019 (1.5), 107.0120 (0.3)	4.65	1.529	2	2
**62**	eriodictyol 7-O-dihexoside	C_27_H_32_O_16_	611.1624	611.16.24 (71.2), 287.0562 (75.1), 267.1056 (0.8), 151.0023 (100), 135.0436 (59.2), 125.0228 (5.7), 107.0121 (23.2)	4.58	0.969	2	1
**63**	luteolin 7-*O*-dihexoside	C_27_H_30_O_16_	609.1468	609.1468 (74.9), 447.0940 (0.3), 285.0402 (100), 284.0326 (7.64), 267.0293 (0.4), 256.0377 (0.9), 241.0497 (0.8), 217.0498 (1.3), 199.0393 (2.1), 151.0024 (3.7), 133.0280 (5.1), 107.0122 (2.3)	4.72	1.120	1,2	1
**64**	luteolin 7-*O*-hexosyl-*O*-hexuronide	C_27_H_28_O_17_	623.1260	623.1260 (64.1), 461.0724 (6.4), 285.0403 (100), 284.0320 (2.42), 256.0376 (0.8), 241.0497 (0.8), 229.0495 (0.2), 217.0498 (1.2), 199.0392 (2.0), 151.0023 (1.9), 133.0278 (5.5), 107.0123 (1.9)	4.84	1.071	1	2
**65**	luteolin 7-*O*-pentosyl-*O*-hexoside	C_26_H_28_O_15_	579.1302	579.1302 (74.6), 447.0934 (0.3), 285.0403 (100), 256.0364 (1.7), 227.0339 (0.9), 217.0500 (1.3), 151.0023 (4.0), 133.0279 (4.7), 107.0124 (2.0)	5.08	1.082	1,2	2
**66**	rutin ^a^	C_27_H_30_O_16_	609.1461	609.1469 (100), 301.0347 (37.2), 300.0276 (62.2), 271.0248 (35.6), 255.0296 (16.8), 243.0291 (7.1), 227.0353 (1.9), 211.0404 (0.7), 199.0384 (0.5), 178.9982 (3.1), 163.0024 (1.0), 151.0025 (5.2), 121.0276 (0.5), 107.0122 (1.6)	5.09	1.317	1,2	1
**67**	isoquercitrin ^a^	C_21_H_20_O_12_	463.0884	463.0884 (100), 301.0343 (36.7), 300.0273 (81.5), 271.0249 (31.4), 227.0342 (0.89), 178.9974 (2.63), 243.0291 (6.57), 151.0022 (7.56), 107.0120 (3.47)	5.18	0.455	1	1
**68**	luteolin 7-O-rutinoside ^a^	C_27_H_30_O_15_	593.1520	593.1520 (84.1), 285.0405 (100), 284.0331 (11.61), 256.0371 (1.2), 241.0522 (0.6), 217.0493 (0.6), 199.0389 (1.1), 151.0021 (3.4), 133.0279 (4.9), 107.0123 (2.0)	5.22	1.428	1,2	1
**69**	eriodictyol-7-*O*-hexoside	C_21_H_22_O_11_	449.1086	449.1086 (15.3), 287.0560 (100), 151.0023 (66.2), 135.0437 (52.9), 125.0229 (3.8), 107.0122 (13.7)	5.26	0.856	2	2
**70**	eriodictyol-7-*O*-hexuronide	C_28_H_16_O_6_	463.0830	463.0830 (72.0), 287.0471 (40.2), 286.0437 (100), 151.0023 (33.0), 135.0436 (20.4), 125.0224 (1.7)	5.33	1.447	1	2
**71**	luteolin 7-*O*-hexuronide	C_21_H_18_O_12_	461.0731	461.0731 (100), 285.0403 (100), 267.0289 (0.3), 241.0501 (0.8), 229.0480 (0.2), 217.0503 (1.1), 151.0024 (4.7), 133.0280 (9.5), 107.0122 (2.5)	5.35	1.173	1	2
**72**	luteolin 7-*O*-glucoside ^a^	C_21_H_20_O_11_	447.0933	447.0933 (100), 285.0401 (86.0), 284.0325 (35.24), 256.0374 (3.9), 239.0343 (0.6), 227.0343 (2.1), 211.0391 (1.1), 151.0023 (5.0), 133.0279 (4.1), 107.0123 (2.9)	5.39	0.012	1,2	1
**73**	apigenin 4′-*O*-hexoside	C_21_H_20_O_10_	431.0987	431.0987 (83.9), 269.0455 (100), 225.0548 (4.4), 151.0024 (3.5), 117.0330 (9.3), 107.0122 (4.9)	5.44	0.673	2	2
**74**	chrysoeriol 4′-*O*-dihexoside	C_28_H_32_O_16_	623.1628	623.1628 (32.7), 299.0559 (100), 298.0479 (0.2), 284.0326 (43.0), 227.0339 (0.6), 151.0022 (1.3), 133.0279 (0.2), 107.0119 (0.6)	5.47	1.640	2	2
**75**	quercetin 3-*O*-acetylhexoside	C_23_H_22_O_13_	505.0992	505.0992 (5.9), 463.0870 (1.1), 301.0344 (33.9), 300.0274 (87.0), 271.0247 (42.5), 255.0294 (18.5), 243.0298 (7.7), 227.0334 (1.4), 178.9976 (28.3), 163.0028 (2.4), 151.0019 (5.4), 107.0123 (1.2)	5.61	0.765	1,2	2
**76**	isoetin 4′-*O*-hexoside	C_21_H_20_O_12_	463.0886	463.0886 (96.52), 301.0353 (100), 300.0271 (7.99), 151.0024 (16.76), 149.0231 (30.03), 107.0122 (10.60)	5.63	0.455	1	3
**77**	apigenin 7-*O*-rutinoside	C_27_H_30_O_14_	577.1572	577.1572 (33.6), 269.0454 (100), 151.0021 (1.0), 149.0231 (0.9), 117.0330 (3.7), 107.0125 (1.7)	5.81	1.648	2	1
**78**	isorhamnetin 3-*O*-glucoside ^a^	C_22_H_22_O_12_	477.1042	477.1040 (100), 315.0504 (11.2), 314.0434 (59.5), 299.0210 (3.5), 271.0248 (16.6), 257.0447 (4.8), 243.0293 (25.7), 227.0346 (3.9), 215.0343 (3.6), 199.0390 (1.8), 151.0021 (1.7), 135.0435 (5.5)	5.92	0.253	1,2	1
**79**	apigenin 7-*O*-glucoside ^a^	C_21_H_20_O_10_	431.0982	431.0981 (100), 268.0375 (65.3), 269.0447 (28.6), 211.0394 (1.9), 151.0023 (4.1), 117.0329 (1.8), 107.0123 (2.4)	6.04	0.831	1,2	1
**80**	luteolin 7-*O*-glucoside ^a^	C_21_H_20_O_11_	447.0933	447.0933 (23.7), 285.0403 (100), 284.0334 (1.1), 211.1346 (0.2), 151.0024 (5.2), 133.0280 (10.0), 107.0123 (2.4)	6.05	0.012	1,2	1
**81**	apigenin 7-*O*-hexuronide	C_21_H_18_O_11_	445.0778	445.0778 (27.5), 269.0454 (100), 225.0546 (1.6), 175.0236 (14.9), 151.0022 (1.8), 117.0331 (7.2), 107.0122 (2.7)	6.10	0.372	1,2	2
**82**	luteolin 7-O-acetylhexoside	C_23_H_22_O_12_	489.1043	489.1043 (100), 447.1004 (0.3), 327.0506 (1.1), 285.0403 (70.2), 284.0326 (36.2), 227.0343 (1.9), 151.0024 (4.0), 133.0280 (3.1), 107.0125 (2.8)	6.23	1.003	1,2	2
**83**	luteolin 7-*O*-dihydroxybutyryl-*O*-acetylhexoside	C_27_H_28_O_15_	591.1364	591.1364 (100), 529.1375 (12.9), 489.1044 (37.9), 447.0934 (28.0), 327.0497 (1.0), 285.0404 (91.1), 284.0326 (30.3), 227.0340 (1.0), 151.0025 (4.5), 133.0282 (6.4), 107.0121 (1.2)	6.26	1.365	1,2	2
**84**	isorhamnetin 3-*O*-acetylhexoside	C_24_H_24_O_13_	519.1149	519.1149 (100), 387.1098 (0.8), 357.0608 (0.7), 315.0509 (47.5), 314.0434 (58.8), 271.1098 (0.8), 357.0608 (0.7), 315.0509 (47.5), 314.0434 (58.8), 271.0248 (25.7), 257.0454 (2.3), 243.0295 (21.2), 227.0347 (4.9), 151.0025 (4.0), 133.0281 (2.9), 135.0433 (0.2), 107.0123 (1.1)	6.46	0.994	1,2	1
**85**	isoetin	C_15_H_10_O_7_	301.0352	301.0352 (100), 271.0245 (0.3), 255.0290 (0.9), 151.0022 (4.3), 149.0231 (21.3), 137.0230 (0.2), 133.0279 (0.3), 121.0278 (0.9), 107.0214 (3.3)	6.48	−0.418	2	3
**86**	chrysoeriol-4′-*O*-hexuronide	C_23_H_22_O_12_	475.0882	475.0888 (100), 299.0558 (27.6), 284.0325 (45.1), 256.0376 (5.2), 227.0347 (1.6), 199.0385 (0.9), 151.0024 (4.9), 133.0278 (4.7), 107.0120 (3.4),	6.75	1.265	1,2	2
**87**	apigenin 7-*O*-acetylhexoside	C_23_H_22_O_11_	473.1093	473.1093 (100), 413.0882 (3.4), 311.0438 (3.1), 297.0413 (1.3), 269.0448 (20.8), 268.0377 (52.6), 211.0392 (1.4), 151.0023 (3.9), 117.0330 (1.9), 107.0123 (2.7)	6.76	0.730	2	2
**88**	apigenin 7-*O*-dihydroxybutyryl-*O*-acetylhexoside	C_27_H_28_O_14_	575.1411	575.1411 (45.6), 513.1402 (12.8), 473.1091 (40.5), 431.0986 (40.8), 413.0847 (0.4), 311.0564 (0.8), 269.0453 (100), 268.0376 (32.7), 151.0021 (2.4), 117.0031 (4.8), 107.0121 (3.2)	6.93	0.837	2	2
**89**	luteolin 7-O-acetylhexoside isomer	C_23_H_22_O_12_	489.1041	489.1041 (100), 447.0960 (0.8), 327.0508 (1.3), 285.0401 (73.7), 284.0325 (38.1), 227.0344 (2.1), 151.0023 (4.9), 133.0280 (3.7), 107.0122 (3.2)	7.05	0.431	1,2	2
**90**	eriodyctiol ^a^	C_15_H_12_O_6_	287.0561	287.0563 (18.5), 151.0023 (100), 135.0437 (89.3), 125.0228 (5.1), 109.0277 (1.8), 107.0123 (12.5)	7.42	−0.074	1,2	1
**91**	luteolin ^a^	C_15_H_10_O_6_	285.0405	285.0402 (100), 257.0457 (0.3), 241.0984 (0.9), 229.0490 (0.2), 217.0498 (1.1), 199.0391 (1.7), 151.0023 (4.9), 133.0280 (23.3), 107.0123 (4.0)	7.57	−1.057	1,2	1
**92**	quercetin ^a^	C_15_H_10_O_7_	301.0354	301.0354 (100), 257.0477 (22.1), 178.9975 (17.6), 151.0023 (36.7), 121.279 (9.4), 107.0122 (9.5)	7.63	0.080	2	1
**93**	apigenin ^a^	C_15_H_10_O_5_	269.0457	269.0453 (100), 225.0549 (0.5), 201.0545 (0.8), 151.0023 (6.4), 117.0330 (20.4), 107.0122 (5.3)	8.62	−0.917	1,2	1
**94**	cirsiliol	C_17_H_14_O_7_	329.0667	329.0667 (100), 314.0434 (60.3), 299.0197 (32.3), 300.0237 (2.8), 271.0247 (25.5), 243.296 (4.8), 199.0399 (1.9), 161.0231 (11.5), 151.0024 (2.0), 133.0279 (0.3), 107.0120 (0.2)	8.87	0.012	2	1
**95**	chrysoeriol	C_16_H_12_O_6_	299.0551	299.0551 (100), 285.0347 (7.3), 284.0325 (81.1), 256.0379 (18.0), 151.0023 (4.6)	8.91	−3.449	1,2	2
**Sesquiterpene lactones and derivatives**								
**№**	**Tentatively Annotated Compound**	**Molecular Formula**	**Exact Mass** **[M + H]^+^**	**Fragmentation Pattern in (+) ESI-MS/MS**	**t_R_** **(min)**	**Δ ppm**	**Distribution**	**Level of Identification [28]**
**96**	11β,13-dihydro-15-glucopyranosyllactucin	C_21_H_28_O_10_	441.1745	441.1745 (28.0), 279.1222 (100), 261.1116 (22.0), 243.1011 (12.2), 233.1168 (5.6), 215.1064 (30.6), 197.0959 (5.9), 187.1115 (13.9), 169.1018 (8.7), 159.0804 (19.7), 131.0856 (12.6), 105.0703 (4.7), 91.0548 (4.6), 81.0341 (7.7), 79.0549 (1.2)	3.12	−2.388	1,2	2
**97**	11β,13-dihydrolactucin	C_15_H_18_O_5_	279.1219	279.1219 (54.9), 261.1115 (20.2), 243.1011 (36.5), 233.1168 (23.8), 215.1064 (100), 197.0958 (27.0), 187.1114 (50.3), 169.1010 (29.9), 159.0803 (82.4), 131.0855 (45.9), 105.0702 (19.8), 91.0547 (20.7), 81.0705 (3.3), 79.0548 (5.8)	3.93	−2.795	1,2	2
**98**	lactucin	C_15_H_16_O_5_	277.1069	277.1069 (57.5), 259.0959 (25.7), 241.0855 (45.1), 231.1010 (22.1), 213.0907 (100), 195.0804 (35.8), 185.0960 (80.1), 167.0854 (38.6), 157.1012 (16.9), 142.0778 (32.5), 129.0701 (30.8), 109.0286 (3.7), 91.0547 (29.3), 81.0342 (3.4), 79.0549 (9.1)	4.86	−0.650	1,2	2
**99**	15-hydroxytaraxacin	C_15_H_14_O_4_	259.0959	259.0959 (100), 241.0855 (43.6), 231.1012 (30.5), 213.0907 (91.2), 195.0802 (32.9), 185.0959 (69.1), 167.0853 (28.5), 157.1011 (9.1), 142.0777 (36.8), 129.0701 (21.6), 109.0287 (10.9), 79.0549 (5.4)	5.74	−2.298	1,2	2
**100**	8-acetyl-15β-D glucopyranosyllactucin ^a^	C_23_H_28_O_11_	481.1699	482.2729 (2.7),319.1171 (100), 259.0963 (79.327), 241.0854 (16.5), 213.0909 (35.5), 203.1062 (2.7), 185.0960 (17.8), 167.0852 (10.7), 157.1012 (2.8), 129.0701 (7.7), 97.5018 (13.0), 85.0288 (23.9)	5.94	−1.077	1,2	1
**101**	8-deoxylactucin	C_15_H_16_O_4_	261.1115	261.1115 (51.4), 243.1011 (32.8), 225.0905 (20.7), 215.1064 (100), 197.0959 (43.9), 187.0753 (80.1), 169.1010 (46.9), 159.0804 (66.0), 131.0856 (45.1), 121.0285 (20.5), 91.0547 (17.7), 81.0706 (1.1)	6.09	−2.625	1,2	2
**102**	8-acetyl-11β,13-dihydro-15-glucopyranosyllactucin	C_23_H_30_O_11_	483.1853	483.2204 (9.7), 321.1326 (100), 261.1114 (72.2), 243.1009 (37.2), 215.1064 (32.9), 205.0860 (5.8), 187.0755 (15.9), 169.1006 (11.8), 159.0806 (17.8), 131.0858 (12.5), 105.0699 (2.6), 91.0547 (4.9), 81.0343 (5.5)	6.12	−1.631	1,2	2
**103**	8-acetyl-11β,13-dihydrolactucin	C_17_H_21_O_6_	321.1329	321.1358 (3.4), 261.1116 (100), 243.1011 (11.7), 233.1170 (3.4), 215.1064 (39.2), 205.0860 (12.1), 187.1114 (16.4), 169.1011 (16.9), 159.0803 (21.9), 131.0856 (17.6), 105.0702 (6.1), 91.0548 (6.1), 81.0708 (1.0), 79.0547 (1.1)	6.25	−1.229	1,2	2
**104**	8-acetyllactucin	C_17_H_18_O_6_	319.1170	319.1170 (2.6), 259.0961 (100), 241.0855 (23.7), 231.1013 (11.0), 213.0907 (56.1), 195.0802 (14.6), 185.0960 (39.6), 167.0854 (13.7), 157.1012 (14.1), 142.0777 (14.0), 129.0700 (11.9), 109.0287 (21.0), 79.0550 (4.4), 81.0341 (7.3), 91.0548 (6.1)	6.99	−2.020	1,2	2
**105**	sonchuside A ^a^	C_21_H_32_O_8_	411.2026 [M − H]^−^	457.2084 (+HCHO) (100), 411.2026 (30.9), 249.1495 (28.8), 205.1596 (1.1), 145.0602 (3.7), 127.0496 (5.2)	7.85	0.484	1.2	1
**Coumarins**								
**106**	aesculin	C_15_H_16_O_9_	341.0863	341.0863 (7.6), 179.0336 (100), 151.0390 (2.6), 135.0437 (1.9), 133.0284 (7.9), 123.0441 (9.3), 85.0289 (2.2), 69.0341 (0.4)	2.71	−0.222	1,2	2
**107**	7-hydoxycoumarin (umbelliferone)	C_9_H_6_O_3_	163.0388	163.0388 (100), 145.0283 (40.7), 135.0440 (64.7), 121.0650 (1.7), 117.0337 (40.6), 107.0495 (8.9), 89.0391 (48.2), 79.0549 (6.8)	3.16	−1.230	1,2	2
**108**	aesculetin	C_9_H_6_O_4_	179.0337	179.0337 (100), 151.0389 (5.9), 133.0284 (15.2), 123.0442 (24.6), 105.0701 (2.8), 91.0548 (1.5)	3.47	−0.979	1,2	2
**109**	coumarin	C_9_H_6_O_2_	147.0440	147.0440 (49.4), 119.0493 (100), 91.0547 (85.2), 65.0394 (12.1), 53.0394 (0.4)	3.94	−0.517	1,2	2
**110**	ostruthol	C_21_H_22_O_7_	387.1431	387.1431 (66.9), 369.1321 (57.5), 351.1218 (16.5), 299.0544 (11.6), 233.0806 (6.7), 203.0697 (6.8), 191.0701 (8.1), 165.0545 (100), 166.0577 (2.7), 151.0386 (7.6), 137.0397 (48.1), 114.0916 (12.6), 91.0547 (2.7), 79.0549 (7.2)	6.52	−1.962	1,2	2

^a^—compare to reference standard; 1—*C. alpina* leaf extract; 2—*C. alpine* flowering heads extract.

Compounds **58**, **60**, and **76** were identified as glycosides of isoetin (**85**). Isoetin is a flavone, an isobaric compound of quercetin with [M−H]^−^ at *m*/*z* 301.035, with an additional hydroxyl group in the B ring. The structure could be evidenced by the presence of more intense RDA fragment ion ^1,3^B^−^ at *m*/*z* 149.023 (21.32%) compared to ^1,3^A^−^ at *m*/*z* 151.003 (4.27%) (Figure 2). The ^1,3^B^−^ ion is typical for flavones, while ^1,2^A^−^ could be found in the MS/MS spectrum of the flavanols. The flavone isoetin could be distinguished from the flavonol quercetin by the presence of fragment ^1,3^B^−^ at *m*/*z* 149.023, while ^1,2^A^−^ at *m*/*z* 178.997 is characteristic of quercetin and its glycosides [27].

Compounds **74** and **86** were identified as glycosides of chrysoeriol (95) (*m*/*z* at 299.055) based on the MS/MS spectra with diagnostic fragment ions resulting from the successive loss of methyl radical •CH_3_ (at *m*/*z* 284.032), CO (at *m*/*z* 256.037) and CHO (at *m*/*z* 227.034), as well as RDA ions ^1,3^A^−^ at *m*/*z* 151.003, ^0,4^A^−^ at *m*/*z* 107.012, and ^1,3^B^−^ at *m*/*z* 133.028. (Table 1). Compound 94 ([M−H]^−^ at *m*/*z* 329.066) showed fragment ions at *m*/*z* 314.043 [M−H-•CH_3_]-, 299.019 [M−H-2•CH_3_]^−^, 271.025 [M−H-2•CH_3_-CO]^−^, 227.035 [M−H-2•CH_3_-CO-CO_2_]-, as well as RDA fragment ions at *m*/*z* 161.023 (^1,3^A^−^-CH_4_-H_2_O), 133.027 (^1,3^B^−^) 151.002 (^1,3^A^−^-CH_4_-CO) (Table 1, Appendix A). Compound **94** was identified as cirsiliol [19].

##### Flavonols

Compounds **56–57**, **66–67**, and **75** revealed losses of hexosyl-hexuronic acid, dihexose, rutinose, hexose, and acetylhexose, respectively; the aglycone was recorded at *m*/*z* 301.034 and corresponded to quercetin (**92**). As a result of RDA cleavage of C-ring, fragment ions ^1,3^A^−^ at *m*/*z* 151.002, ^0,4^A^−^ at *m*/*z* 107.012, ^1,2^A^−^ at *m*/*z* 179.998, and ^1,2^B^−^ at *m*/*z* 121.028 were formed (Table 1, Appendix A). Similarly, **61**, **78**, and **84** were identified as glycosides of isorhamnetin [19].

##### Flavanones

In the MS/MS spectra of **62**, **69**, and **70**, loss of dihexose, hexose, and hexuronic acid, respectively, was observed; the aglycone was recorded at *m*/*z* 287.056 corresponding to eriodictyol [27]. Key fragments in the identification of these compounds were the RDA ions ^1,3^B^−^ at *m*/*z* 135.043 and ^1,4^A^−^ at *m*/*z* 125.022 (Table 1, Appendix A).

Depending on the intensity and the ratio of the fragment ions [Y0]^−^ and [Y0-H]^−^, the sites for binding the sugar parts to the aglycones were also determined [29]. The identification of compounds **66**–**68**, **72**, **78**–**80**, and **90**–**93** was confirmed by comparison with reference standards.

##### Sesquiterpene Lactones (STLs)

STL dereplication is based on fragmentation patterns and diagnostic ions in the positive ionization mode as more informative for this class of specialized metabolites [19,30]. Based on accurate mass MS spectra, MS/MS fragmentation, relative intensities of precursor and fragment ions, and elemental composition, nine guanolide STLs (**96**–**104**), derivatives of lactucin and dihydrolactucin, were tentatively identified in *C. alpina* extracts. Among them, three are glycosylated (**96**, **100**, and **102**), and three are acetylated (**102–104**). MS/MS fragmentation of terpenes, including the presence of characteristic ions corresponding to the loss of H_2_O (−18 Da), 2H_2_O (−36 Da), CO (−28 Da), CO_2_ (−44 Da), CH_3_COOH (−60 Da), as well as accompanying loss of H_2_O + CO (−46 Da), 2H_2_O+CO (−64 Da), H_2_O+CO_2_ (−62 Da) (Appendix A). In addition, in negative ion a germacranolide (**105**), sonchuside A was identified in *C. alpina* leaves extract (Table 1). The identification of compounds **100** and **105** were confirmed by comparison with reference standards [8].

##### Coumarins

Compound **109** ([M+H]^+^ at *m/z* 147.044) gave a base peak at *m*/*z* 119.049 and an intense ion at *m*/*z* 91.054 (85.17%), resulting from the sequential loss of two CO groups. Thus, the coumarin structure was proposed for **109** [31]. A similar MS/MS spectrum was obtained for **107**, but here an initial loss of H_2_O was observed, resulting from the loss of the OH group, and the compound was identified as 7-hydroxycoumarin (umbelliferon) (Table 1). By analogy, but with two hydroxyl groups, the fragmentation patterns of aesculetin (**108**) and aesculin (**106**) are explained (Table 1, Appendix A) [31].

### 2.2. Total Phenolic Compounds and Flavonoids Content; Antioxidant and Enzyme Inhibitory Activity

Regarding the content of total phenolic compounds, the leaves showed a higher content (75.13 ± 0.51 mg GAE/g), while a higher content of total flavonoids was observed in the flowering heads of *C. alpina* (Table 2). 

Various tests were performed to determine the antioxidant profile of the plant extracts. Tests based on different mechanisms have been used in the current work. The results are presented in Table 2. *C. alpina* leaves showed higher antioxidant activity in all of the used methods. The DPPH radical scavenging activity of the leaves extract was 132.80 ± 3.77 mg TE/g, and for ABTS, the value was found to be 139.54 ± 0.57 mg TE/g (Table 2). The reducing capacity of the extracts was evaluated by CUPRAC and FRAP experiments (Table 2). The CUPRAC method evaluated the conversion of Cu (II) to Cu (I), and FRAP indicates the reducing potential of the antioxidant, which reacts with the colorless TPTZ/Fe3^+^ complex to form the blue-colored TPTZ/Fe^2+^. The leaf extract has a high reducing potential (CUPRAC: 212.93 ± 11.59 mg TE/g and FRAP 141.12 ± 6.64 mg TE/g).

One of the most important mechanisms of action of antioxidants is the chelation of pro-oxidant metals. Iron is the most active metal that causes oxidative changes in cells, mainly proteins and lipids. Table 2 presented the total antioxidant activity of the extracts, assessed by the phospho-molybdenum method and the metal-chelating ability. Again, the leaves exhibited the highest total antioxidant activity (1.55 ± 0.04 mmol TE/g) and metal-chelating ability (36.97 ± 0.51 mg EDTAE/g).

The enzyme inhibitory activity of the studied extracts was determined against acetyl- and butyrylcholinesterase, α-amylase, α-glucosidase, and tyrosinase (Table 3). 

*C. alpina* leaves extract showed higher acetylcholinesterase (1.98 ± 0.02 mg GALAE/g) and butyrylcholinesterase inhibitory activity (0.74 ± 0.06 mg GALAE/g) (Table 3). Flowering heads showed no butyrylcholinesterase activity. Both enzymes are considered therapeutic targets in the treatment of Alzheimer’s disease. The *C. alpina* leaves extract also showed high activity against the enzyme tyrosinase (49.87 ± 3.19 mg KAE/g) (Table 3). This enzyme plays a key role in the biosynthesis of melanin, being responsible for skin pigmentation. Increased melanin formation leads to skin diseases such as hyperpigmentation, skin spots, etc. Tyrosinase inhibitors are becoming increasingly important hypopigmenting agents in cosmetic and medicinal products. Regarding α-amylase and α-glucosidase inhibitory effects, the *C. alpina* flowering heads extract (amylase: 0.47 mmol ACAE/g and glucosidase: 1.05 mmol ACAE/g) was more active on both enzymes than leaves extract (amylase: 0.28 mmol ACAE/g and glucosidase: 0.60 mmol ACAE/g). Inhibition of these enzymes is known to be an important therapeutic strategy to control blood glucose levels in diabetic patients after a carbohydrate-rich diet. In this sense, the tested *C. alpina* parts could be considered as a multifunctional bioactive agent from antioxidants to enzyme inhibitors, and thus, the presented study could be valuable to provide an effective raw material in the pharmaceutical, nutraceutical, and cosmeceutical industries. 

### 2.3. Multivariate Analysis

After the univariate analysis, eleven specialized metabolites were used to generate the PLS-DA model. PLS-DA plot demonstrated significant discrimination of both leaves and flowering heads of *C. alpina* (Figure 3A). A point to note is that there is not any overlap between both extracts, and the model has performed a 100% separation (Figure 3B). The best performance of the model was achieved for 1 component. Afterward, the importance of each bioactivity for the generating of the first component was investigated. As suggested by [32], VIPs above 1 are important and have a significant role in this model. Thus, referring to Figure 3C, all bioactivities, except acetylcholinesterase and tyrosinase, have an important role in the discrimination of the leaves and flowering heads of *C. alpina*. Therefore, *C. alpine* leaves appear to be more perspective as a result of their prominent bioactivity (Figure 3D). 

The bioactivities varied significantly within the studied *C. alpina* plant parts due to the presence of different metabolites in each organ responsible for the specific biological function and role in plant development, reproduction, and growth [33]. Besides, to visualize the molecules’ contrast between both plant extracts, a line plot was plotted using the peak area database. Before the graphic representation, the peak area was log2 transformed. There is a great variation in the molecule levels for all the subclasses (Appendix A). Regarding the first subclass (carboxylic, hydroxybenzoic, and hydroxycinnamic acids), salicylic acid (**13**) and syringic acid-*O*-hexoside (**3**) were relatively abundant in the leaves extract, while the level of quinic acid (**11**) was relatively higher in the flowering heads extract. 

Concerning the second subclass (hydroxycinnamic acids and derivatives), rosmarinic acid (**16**) was found only in the leaves. In addition, the leaves were rich in caffeoylcitramalic acid (**22**), while the flowering heads exhibited a relatively high concentration of caffeic acid-*O*-hexoside isomer (**17**) and caffeic acid (**19**). As regards the acylquinic acids subclass, four compounds, including 3-*p*-coumaroylquinic acid (**24**), 1-caffeoyl-3-hydroxydihydrocaffeoylquinic acid (**31**), 1,3,4-tricaffeoylquinic acid (**32**), and 1,3-dicaffeoyl-5-hydroxydihydrocaffeoylquinic acid (**40**) were not presented in the flowering heads extract. However, this extract possessed a relatively high amount of several compounds i.e., neochlorogenic (3-caffeoylquinic) acid (**23**), 5-feruloylquinic acid (**29**), 3,4,5-tricaffeoylquinic acid (**43**), to mention only a few. In contrast, the leaf extract was rich in 1/3/5-caffeoyl-4-hydroxydihydrocaffeoylquinic acid (**30**). Overall, the leaf extract was relatively rich in secondary metabolites belonging to the hydroxycinnamoyltartaric acids subclass. Relating to flavones, flavonols, and flavanones, five compounds are present only in the leaves, while flowering heads have 16. However, among the metabolites presented at once in both plant organs, the flowering heads are richer in several compounds than the leaves, including luteolin 7-O-rutinoside (**68**), quercetin 3-*O*-acetylhexoside (**75**), isorhamnetin 3-*O*-glucoside (**78**), apigenin 7-*O*-glucoside (**79**), luteolin 7-*O*-glucoside (**80**), apigenin 7-*O*-hexuronide (**81**), luteolin 7-O-acetylhexoside (**82**), isorhamnetin 3-*O*-acetylhexoside (**84**), eriodyctiol (**90**), luteolin (**91**), apigenin (**93**) and chrysoeriol (**95**). Concerning the last two subclasses, both parts present practically the same compounds, notwithstanding some differences observed in 8-acetyl-15β-D -glucopyranosyllactucin (**100**), 15-hydroxytaraxacin (**99**), aesculin (**106**). Thus, the flowering heads extract is richer in flavonoids compared to the leaves but contains fewer polyphenols, e.g., acylquinic acids and displays lower bioactivity. Moreover, an antagonistic effect between some of the secondary metabolites might exist. 

Plant polyphenols, e.g., flavonoids and phenolic acids, are multifunctional and can act as reducing agents, hydrogen-donating antioxidants, and singlet oxygen quenchers [15]. Key points in the structure of flavonoids responsible for the antioxidant activity are as follows: the o-dihydroxy structure in the B ring, the 2,3 double bond in conjugation with a 4-oxo function in the C ring, and the 3- and 5-OH groups with 4-oxo function in A and C rings, requiring for maximum radical scavenging potential. Thus, quercetin is satisfied all the above-mentioned determinants and is a more effective antioxidant than the flavanols [15]

Regarding the phenolicacids, it was found that the diphenolics, chlorogenic and caffeic acids, demonstrated higher radical scavenging ability than monophenolics (*p*-coumaric acid), consistent with the chemical criteria applied to diphenolics. Methoxylation of the hydroxyl group in the ortho position of the diphenolic acids, as in ferulic acid, results in a decrease in the scavenging reaction, hydroxylation as in caffeic acid in place of methoxylation is substantially more effective. Ferulic acid is, indeed, expected to be more effective than p-coumaric acid due to the electron-donating methoxy group allowing increased stabilization of the resulting aryloxyl radical through electron delocalization after hydrogen donation by the hydroxyl group [15].

Previous investigation revealed that dicaffeoyl derivatives cichoric acid and 1,5-dicaffeoylquinic acid demonstrated higher DPPH activity compared to monocaffeoyl derivatives, caffeoyltartaric acid (caftaric acid) and chlorogenic acid, respectively [18]. EC50 values for monocaffeoyl derivatives were found to be in the order of 20 μM, while those for dicaffeoyl derivatives had values of about 10 μM [18].

AChE activity of sesquiterpene lactones (lactucin and lactucopicrin) and different chicory extracts was previously determinated using isothermal titration calorimetry (ITC) and docking simulation. The results showed strong interactions of STLs as well as extracts from chicory with AChE. In a test of enzyme activity inhibition after introducing acetylcholine into the model system with STL, a stronger ability to inhibit the hydrolysis of the neurotransmitter was observed for lactucopicrin, which is one of the dominant STL in chicory. The inhibition of enzyme activity was more efficient in the case of extracts [34].

α-Amylase and α-glucosidase inhibitory activities of caffeic and chlorogenic acids in a dose-dependent manner (2–8 μg/mL) were previously evaluated. However, caffeic acid had a significantly higher inhibitory effect on α-amylase with IC50 3.68 μg/mL and α-glucosidase (IC50 = 4.98 μg/mL) than chlorogenic acid (α-amylase IC50 = 9.10 μg/mL and α-glucosidase IC50 = 9.24 μg/mL). Furthermore, both phenolic acids exhibited high antioxidant properties, and caffeic acid demonstrated a higher DPPH effect [35].

The presented study revealed 110 secondary metabolites, including 13 carboxylic, hydroxybenzoic acids, and their glycosides, 9 hydroxycinnamic acids, and derivatives, 21 acylquinic acids, 12 acyltartaric acids, 40 flavones, flavonols, and flavanones, 5 coumarins and 10 sesquiterpene lactones. Ninety-five of all annotated compounds are reported for the first time in *C. alpina*. Our results for the total flavonoid content could be compared to those obtained for the wild collection of alpine chicory by Alexandru et al. [4], while the total phenolic content is significantly higher than their result. Oppositely, the data for the edible shoots of cultivated *C. alpina* are prominently higher compared to our data. 

### 2.4. Cytotoxicity Assay

To evaluate the *C. alpina* cytotoxicity, a human monocytic cell line (THP-1 cells) mimics the behavior of the extracts toward the immune system was used (Figure 4). After 24 h of incubation of macrophage cell line THP-1 with flowering heads and leaves extracts, a slight rise of metabolic activity is measured between 2 and 200 µg.mL^−1^. We reach toxicity of 70% for 2000 and 3000 µg.mL^−1^ for flowerings head extracts, and of almost 90% for leaves, at the same concentration. The latter concentrations are very high and could be considered meaningless.

## 3. Materials and Methods

### 3.1. Plant Material 

*C. alpina* leaves and flowering heads were collected at Vitosha Mt., Bulgaria at 1720 m a.s.l. (42.60° N 23.25° E), during the full flowering stage in August 2022. The plant was identified by one of us (V. B.) according to Stojanov et al. [36]. A voucher specimen was deposited at the Herbarium Academiae Scientiarum Bulgariae (SOM 177 802). Twenty plant samples were separated into leaves and flowering heads and dried at room temperature.

### 3.2. Chemicals

Acetonitrile (hypergrade for LC–MS), formic acid (for LC-MS), and methanol (analytical grade) were purchased from Chromasolv (Bulgaria). The authentic standards gallic, vanillic, protocatechuic, gentisic, salicylic, *p*-coumaric, rutin, isoquercitrin, luteolin 7-*O*-rutinoside, luteolin 7-*O*-glucoside, isorhamnetin 3-*O*-glucoside, apigenin 7-*O*-glucoside, luteolin, quercetin, apigenin, and chrysoeriol were obtained from Extrasynthese (Genay, France). Rosmarinic, caffeic, ferulic, chlorogenic, neochlorogenic, 3,4-dicaffeoylquinic, 3,5-dicaffeoylquinic, 1,5-dicaffeoylquinic, and 4,5-dicaffeoylquinic acid were supplied from Phytolab (Vestenbergsgreuth, Germany). Sonchuside A and 8-acetyl-15β-D -glucopyranosyllactucin were previously isolated and identified by Zidorn et al., 2005 [8].

### 3.3. Sample Extraction

Air-dried powdered leaves (50 g) and flowering heads (10 g) were extracted with 80% MeOH (1:20 *w*/*v*) by sonication (100 kHz, ultra-sound bath Biobase UC-20C) for 15 min (×2) at room temperature. The methanol was evaporated in vacuo, and water residues were lyophilized (lyophilizer Biobase BK-FD10P) to yield crude extracts as follows: leaves—10.67 g and flowering heads—0.90 g. Then, the lyophilized extracts were dissolved in 80% methanol (0.1 mg/mL), filtered through a 0.45 μm syringe filter (Polypure II, Alltech, Lokeren, Belgium), and an aliquot (2 mL) of each solution was subjected to UHPLC–HRMS analyses. The same extracts were used for pharmacological tests.

### 3.4. Ultra-High-Performance Liquid Chromatography–High Resolution Mass Spectrometry (UHPLC-HRMS)

Mass spectrometry analyses were carried out on a Q Exactive Plus mass spectrometer (ThermoFisher Scientific, Inc., Waltham, MA, USA) equipped with heated electrospray ionization (HESI-II) probe (ThermoScientific). The mass spectrometer was operated in negative and positive ESI modes within the *m/z* range from 100 to 1000. The other parameters were as follows: spray voltage 3.5 kV (+) and 2.5 kV (−); sheath gas flow rate 38; auxiliary gas flow rate 12; spare gas flow rate 0; capillary temperature 320 °C; probe heater temperature 320 °C; S-lens RF level 50; scan mode: full MS (resolution 70,000) and MS/MS (17,500). The chromatographic separation was performed on a reversed-phase column Kromasil EternityXT C18 (1.8 µm, 2.1 × 100 mm) at 40 °C. The chromatographic analyses were run using 0.1% formic acid in water (A) and 0.1% formic acid in acetonitrile (B) as a mobile phase. The flow rate was 0.3 mL/min. The run time was 33 min. The following gradient elution program was used: 0–1 min, 0–5% B; 1–20 min, 5–30% B; 20–25 min, 30–50% B; 25–30 min, 50–70% B; 30–33 min, 70–95%; 33–34 min 95–5%B. Equilibration time was 4 min [19]. Data were processed by Xcalibur 4.2 (Thermo Scientific) instrument control/data handling software. Metabolite profiling using MZmine 2 software was applied to the UHPLC–HRMS raw files of the studied *C. alpina* extracts.

### 3.5. Total Phenolic and Flavonoid Content

Total phenols and flavonoids were evaluated as gallic acid (GAE) and rutin (RE) equivalents, respectively, using spectrophotometric methods. The experiments were performed as previously reported [37,38].

### 3.6. Determination of Antioxidant and Enzyme Inhibitory Activities

Extracts antioxidant and enzyme inhibitory effects (0.2–1 mg/mL) were evaluated using spectrophotometric assays. Detailed protocols were reported elsewhere [19,39].

### 3.7. Cell Line and Culture

The human monocytic THP-1 (TIB-202) cell line was obtained from the American Type Culture Collection (ATCC, Manassas, VA, USA). The cells grown in RPMI 1640 were supplemented with 10% fetal bovine serum (FBS), 1% glutamine, 1% penicillin/streptomycin, and 0.5% Amphotericin B. Cells were cultured in a humidified atmosphere at 37 °C, under a 5% CO_2_ atmosphere [21].

### 3.8. Cytotoxicity Assay 

THP-1 cells (at 1.10^5^ cells/mL) cells in RPMI medium (Thermo-Fisher) supplemented with 10% FBS were seeded in each well of a 48-well plate (n = 4). Cells were permitted to adhere for 24 h, and then treated with leaves and flowering heads *C. alpina* extracts in the medium for 24 h. Then 40 µL of WST-1 testing solutions (Sigma-Aldrich, St. Louis, MO, USA) was added to each well and the plate incubated at 37 °C for 2 h. The contents of each well were laid down in 3 wells of a 96-well plate. The absorbances were measured at 350 and 630 nm with an Omega StarLab spectrophotometer (Omega, Ortenberg, Germany).

### 3.9. Statistical Analysis 

In the antioxidant and enzyme inhibitory assays, the values are expressed as mean ± SD of three parallel experiments. In terms of antioxidant and enzyme inhibitory abilities, the student t-test (α = 0.05) was performed to determine differences between the tested extracts. The statistical analysis was performed using XlStat 16.0 software. Clustered image maps (CIM) were used to visualize metabolite variation among the extracts. Prior to CIM analyses, data were normalized and centered. Afterward, supervised partial least-square discriminant analysis (PLS-DA) was performed to discriminate the different parts regarding their biological activities. Then CIM was applied to PLS-DA outcomes to characterize each extract. Lastly, Pearson’s correlation coefficients were calculated to evaluate the relationship between secondary metabolites and biological activities, respectively.

## 4. Conclusions

More than 100 secondary metabolites, including carboxylic, hydroxybenzoic, hydroxycinnamic, acylquinic, and acyltartaric acids, flavones, flavonols, flavanones, sesquiterpene lactones, coumarins, and their derivatives were annotated/dereplicated in the *C. alpina* leaves and flowering heads. Ninety-five, including acylquinic acids, acyltartaric acids, and flavonoids, were reported for the first time in *C. alpina*. Cichoric, caftaric, and chlorogenic acids dominated in the leaves, while apigenin, 3,5-di, and 3,4-dicaffeoylquinic acids dominated in the flowering heads profiling. The connection between the different plant parts and biological activity was performed using multivariate statistical analyses. The pronounced antioxidant activity (DDPH, FRAP, CUPRAC, ABTS, Chelating, and Phosphomolibdenum capacity) and enzyme inhibitory potential against AChE, BChE, tyrosinase, and lipase of the leaves extract could be related to the higher content of total polyphenols and the presence of acyltartaric and monoacylquinic acids compare to flowering heads. The prominent α-glucosidase and α-amylase inhibitory activity of the flowering heads correspond to the higher level of total flavonoids, luteolin, apigenin, and their glycosides. The studied extracts expressed low cytotoxicity towards THP-1 viability. In addition to inducing an antioxidant response, *C. alpina* extracts displayed enzyme inhibitory effects in vitro, which generates interest in the plant as a potential candidate for attenuating metabolic-related disorders. Moreover, this study supports further investigation towards the additional in vivo studies and corroborates the application of *C. alpina* in the pharmaceutical and nutraceutical industries.

## Figures and Tables

**Figure 1 plants-12-01009-f001:**
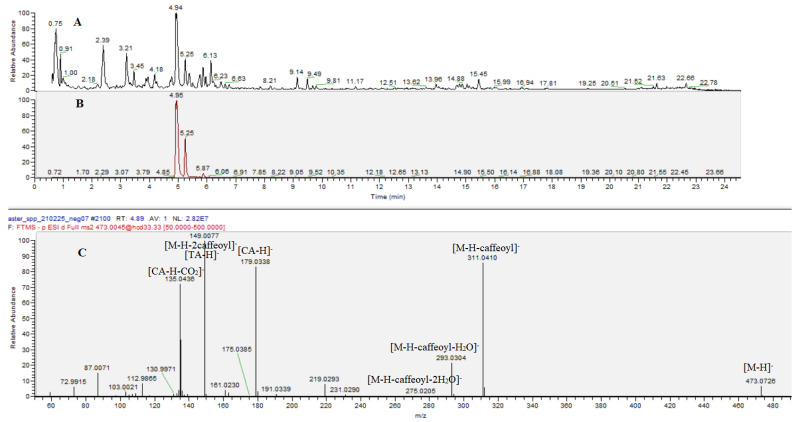
Total ion chromatogram (TIC) in negative ion mode of *C. alpina* leaves extract (**A**); extracted ion chromatogram of peaks at *m/z* 473.072 (**B**); (−) ESI/MS-MS spectrum of dicaffeoyltartaric acid (cichoric acid) (**47**) (**C**).

**Figure 2 plants-12-01009-f002:**
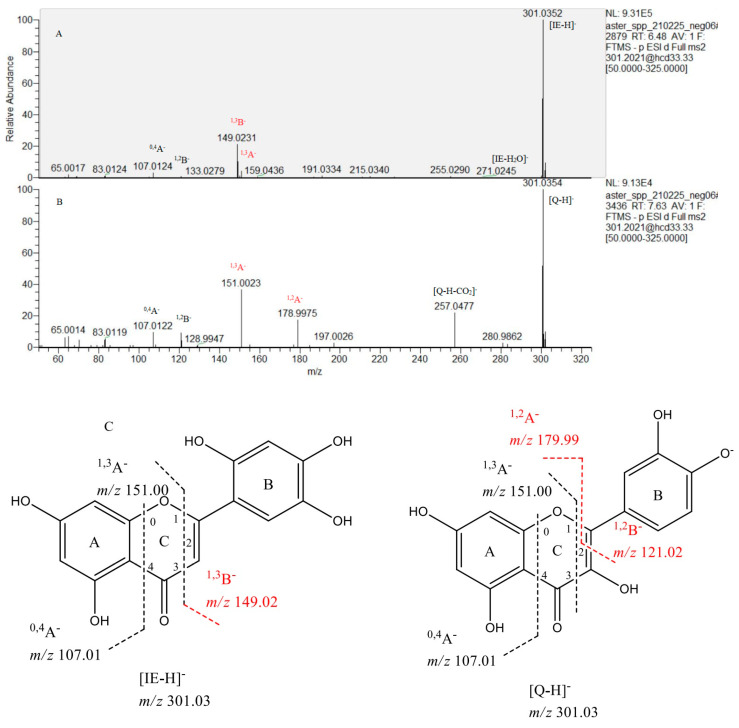
(−) ESI/MS-MS spectrum of isoetin (**85**) (**A**); quercetin (92) (**B**); possible fragmentation and preliminary structure of isoetin (**85**) and quercetin (**92**) (**C**).

**Figure 3 plants-12-01009-f003:**
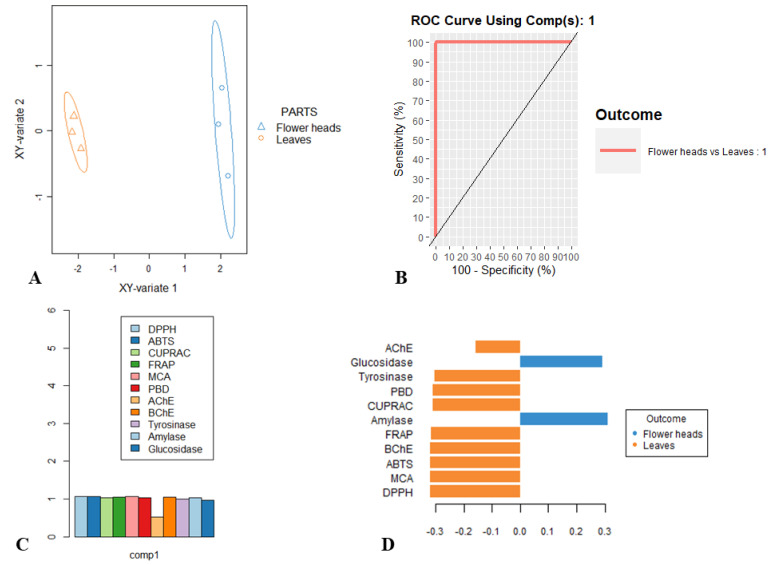
PLS-DA analysis on the bioactivities of *C. alpina.* (**A**) Scatter plot. (**B**) ROC curve. (**C**) VIP score plot. (**D**) Loading plot.

**Figure 4 plants-12-01009-f004:**
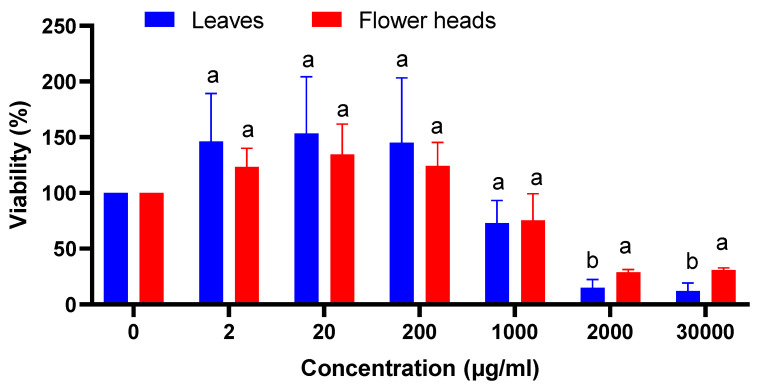
Cytotoxicity of *C. alpina* extracts towards THP-1 cells (by the student t-test, different letters indicate significant difference between plant parts in the same concentration (a and b), *p* ≤ 0.05.).

**Table 2 plants-12-01009-t002:** Total phenolics, and flavonoids content and antioxidant activity of *C. alpina* extracts.

Samples	Total Phenolic Content (mg GAE/g)	Total Flavonoid Content (mg QE)	DPPH (mg TE/g)	ABTS (mg TE/g)	CUPRAC (mg TE/g)	FRAP (mg TE/g)	Chelating (mg EDTAE/g)	Phospho-Molybdenum (mmol TE/g)
Leaves	75.13 ± 0.51 ^a^	17.59 ± 0.10 ^b^	132.80 ± 3.77 ^a^	139.54 ± 0.57 ^a^	212.93 ± 11.59 ^a^	141.12 ± 6.64 ^a^	36.97 ± 0.51 ^a^	1.55 ± 0.04 ^a^
Flowering heads	48.41 ± 0.12 ^b^	27.72 ± 0.36 ^a^	45.36 ± 0.03 ^b^	99.48 ± 1.80 ^b^	157.96 ± 0.96 ^b^	85.97 ± 1.79 ^b^	23.92 ± 0.32 ^b^	1.35 ± 0.02 ^b^

Values are reported as mean ± SD of three parallel experiments. GAE: gallic acid equivalent; QE: quercetin equivalents; TE: Trolox equivalent; EDTAE: EDTA equivalent. Different letters (a and b) indicate significant differences in the tested samples (*p* < 0.05).

**Table 3 plants-12-01009-t003:** Enzyme inhibitory activity of *C. alpina* extracts.

Samples	AChE İnhibition (mg GALAE/g)	BChE İnhibition (mg GALAE/g)	Tyrosinase İnhibition (mg KAE/g)	Amylase İnhibition (mmol ACAE/g)	Glucosidase İnhibition (mmol ACAE/g)	Lipase İnhibition (mg OE/g)
Leaves	1.98 ± 0.02 ^a^	0.74 ± 0.06	49.87 ± 3.19 ^a^	0.28 ± 0.02 ^b^	0.60 ± 0.03 ^b^	4.75 ± 0.21
Flowering heads	1.94 ± 0.06 ^a^	na	33.02 ± 3.88 ^b^	0.47 ± 0.03 ^a^	1.05 ± 0.17 ^a^	na

Values are reported as mean ± SD of three parallel experiments. GALAE: galantamine equivalent; KAE: kojic acid equivalent; ACAE: acarbose equivalent; OE: orlistat equivalent; na: not active. Different letters (a and b) indicate significant differences in the tested samples (*p* < 0.05).

## Data Availability

Not applicable.

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
