# Peer review of "Metabolite profiling and bioactivity of Cicerbita alpina (L.) Wallr. (Asteraceae, Cichorieae)"

_plants, 2023, doi:10.3390/plants12051009_

Round 1

Reviewer 1 Report

In my opinion, the introduction part is too short. It lacks a general introduction which would indicate the purpose of the research. In general, the manuscript is very interesting and the research results obtained are a valuable contribution to the knowledge about the chemical composition of Cicerbita alpina extracts.

Author Response

Response to the Reviewer 1 Comments

In my opinion, the introduction part is too short. It lacks a general introduction which would indicate the purpose of the research. In general, the manuscript is very interesting and the research results obtained are a valuable contribution to the knowledge about the chemical composition of Cicerbita alpina extracts.

Response: Dear reviewer, thanks for the comments. Introduction section was changed according to the recommendation. The purpose of the research was indicated (See page 2, Introduction).

Reviewer 2 Report

The present study describes the phytochemical composition and bioactivity of Cicerbita alpina leaves and flowers. The phytochemical composition is thoroughly presented and discussed. Regarding to the bioactivity, its correlation with specific phytochemicals or group will be interested. Furthermore, please state clearly the novelty of the work.

Author Response

Response to Reviewer 2 Comments

The present study describes the phytochemical composition and bioactivity of Cicerbita alpina leaves and flowers. The phytochemical composition is thoroughly presented and discussed. Regarding to the bioactivity, its correlation with specific phytochemicals or group will be interested. Furthermore, please state clearly the novelty of the work.

Response: Dear reviewer, thanks for the comments. The manuscript was changed according to the recommendation (See page 5, Results and discussion). The purpose of the research was indicated and the novelty of the work was state clearly (See page 2, Introduction and page 25, Conclusion).

Reviewer 3 Report

The article by Zheleva-Dimitrova and co-workers investigated the metabolites of Cicerbita alpina, as well as some in vitro activities. The metabolites investigation was done by dereplication and injection of standards, using LC-MS/MS. This part is very appropriate: the authors report a complete table informing the compounds putatively annotated, molecular formula, exact mass, fragmentation pattern, retention time, error and level of identification. They also report their mass spectra interpretation. I believe the methods are well-performed and the paper is well written. It deserves the publication in Plants.

I just want to raise one point: I don’t believe that multivariate analysis is adding any relevant information. Even the conclusion of this analysis is “paradoxical”. I recommend the removal of this topic.

Author Response

Response to Reviewer 3 Comments

The article by Zheleva-Dimitrova and co-workers investigated the metabolites of Cicerbita alpina, as well as some in vitro activities. The metabolites investigation was done by dereplication and injection of standards, using LC-MS/MS. This part is very appropriate: the authors report a complete table informing the compounds putatively annotated, molecular formula, exact mass, fragmentation pattern, retention time, error and level of identification. They also report their mass spectra interpretation. I believe the methods are well-performed and the paper is well written. It deserves the publication in Plants.

I just want to raise one point: I don’t believe that multivariate analysis is adding any relevant information. Even the conclusion of this analysis is “paradoxical”. I recommend the removal of this topic.

Response: Thank you for your comments. However, we disagree with your comments on parts of the multivariate analysis. Multivariate analysis has gained interest in recent years to establish connections with different parameters. In the literature, this analysis has become popular in the phytochemical studies and more information has been provided the connections between chemical profiles and biological properties in particular. As you can see from Table 1, 110 compounds were identified in the tested extracts, and we have summarized the tables by multivariate analysis in Figure S8 (based on peak area). In addition, we performed various antioxidant and enzyme inhibition studies and Figure 3 clearly showed that the abilities changed the plant parts used. In our study, from Figure 3C, all bioactivities, except acetylcholinesterase and tyrosinase, played important role in the discrimination of the leaves and flowerings heads of C. alpina. Although we found more compounds in the flower head extract, the leaves were more active than the flower head. Although these results appear contradictory, this observation could be explained by the complex nature of phytochemicals or their interactions, such as antagonistic. Similar approaches were taken in our previous studies, and they are listed below.

1- Gevrenova, Reneta, Dimitrina Zheleva-Dimitrova, Vessela Balabanova, Yulian Voynikov, Kouadio Ibrahime Sinan, Mohamad Fawzi Mahomoodally, and Gökhan Zengin. "Integrated phytochemistry, bio-functional potential and multivariate analysis of Tanacetum macrophyllum (Waldst. & Kit.) Sch. Bip. and Telekia speciosa (Schreb.) Baumg.(Asteraceae)." Industrial Crops and Products 155 (2020): 112817.

2- Gevrenova, R., Zengin, G., Sinan, K.I., Zheleva-Dimitrova, D., Balabanova, V., Kolmayer, M., Voynikov, Y. and Joubert, O., 2023. An In-Depth Study of Metabolite Profile and Biological Potential of Tanacetum balsamita L.(Costmary). Plants, 12(1), p.22.

3- Zheleva-Dimitrova, Dimitrina, Gokhan Zengin, Gunes Ak, Kouadio Ibrahime Sinan, Mohamad Fawzi Mahomoodally, Reneta Gevrenova, Vessela Balabanova, Alexandra Stefanova, Paraskev Nedialkov, and Yulian Voynikov. "Innovative biochemometric approach to the metabolite and biological profiling of the Balkan thistle (Cirsium appendiculatum Griseb.), Asteraceae." Plants 10, no. 10 (2021): 2046.

4-Trifan, Adriana, Gokhan Zengin, Kouadio Ibrahime Sinan, Evelyn Wolfram, Krystyna Skalicka-Woźniak, and Simon Vlad Luca. "LC-HRMS/MS phytochemical profiling of Symphytum officinale L. and Anchusa ochroleuca M. Bieb.(Boraginaceae): Unveiling their multi-biological potential via an integrated approach." Journal of Pharmaceutical and Biomedical Analysis 204 (2021): 114283.

5- Ruiz-Riaguas, A., G. Zengin, K. I. Sinan, C. Salazar-Mendías, and E. J. Llorent-Martínez. "Phenolic profile, antioxidant activity, and enzyme inhibitory properties of Limonium delicatulum (Girard) Kuntze and Limonium quesadense Erben." Journal of Chemistry 2020 (2020): 1-10.

Given the above information, please accept our humble request to keep the section in the submitted work.

Reviewer 4 Report

This work is very informative about the identification of compounds.

Firstly I think this article is better situated in other journals like molecules. Most of the work is related to the identification of compounds, where the methods related to plants like vegetative growth. 

Even though the authors wrote many identified compounds and in Table 2 they measured the Total phenolics, flavonoids content, and antioxidant activity of C. alpina extracts, I didn't find the bioactivity of at least the main compounds.

Author Response

Response to Reviewer 4 Comments

This work is very informative about the identification of compounds.

Response: Thank you for your comment.

Firstly I think this article is better situated in other journals like molecules. Most of the work is related to the identification of compounds, where the methods related to plants like vegetative growth. 

Response: Thank you for your comments. The presented manuscript was submitted as an article in Plants, section “Phytochemistry”, Special issue: „Valuable Sources of Bioactive Natural Products from Plants “. This Special Issue of Plants “highlighted the exploration of medicinal plants used in different regions of the world as valuable sources of bioactive compounds. Presently, numerous research groups are focusing on the study and characterization of herbal extracts to identify a variety of plant metabolites and their potential to exert protective effects on biological systems. With the development of modern metabolite profiling methods and metabolomics, a holistic survey of natural extracts and valuable information on their chemical composition has become feasible”.

(https://www.mdpi.com/journal/plants/special_issues/BZ7VCMR635).

In our opinion the presented manuscript is appropriated for journal “Plants”.

Even though the authors wrote many identified compounds and in Table 2 they measured the Total phenolics, flavonoids content, and antioxidant activity of C. alpina extracts, I didn't find the bioactivity of at least the main compounds.

Response: Thank you for your comments. The aim of the presented study was the evaluation of antioxidant and enzyme inhibitory potential of C. alpina leaves and flowering heads methanol-aqueous extracts. In addition, the metabolite profiling together with determination of total polyphenols and flavonoids contents were performed. The isolation of individual compounds was not an object of this study. Several bioactivities of the main compounds were cited in the manuscript according to the reviewer recommendation (See row 396-414).

Round 2

Reviewer 4 Report

As it will be acceptable in Plants, section “Phytochemistry”, Special issue: „Valuable Sources of Bioactive Natural Products from Plants“.

I'm suggesting accepting the article, as the work is very interesting in the isolation and identification of bioactive compounds.